# RETRIEVAL IS ACCURATE GENERATION

**Bowen Cao**[♠,∗] **Deng Cai**[♡,†] **Leyang Cui**[♡] **Xuxin Cheng**[♠] **Wei Bi**[♡] **Yuexian Zou**[♠] **Shuming Shi**[♡]
♠ School of ECE, Peking University
♡ Tencent AI Lab
{cbw2021,chengxx}@stu.pku.edu.cn, zouyx@pku.edu.cn
thisisjcykcd@gmail.com, {leyangcui,victoriabi,shumingshi}@tencent.com

## ABSTRACT

Standard language models generate text by selecting tokens from a fixed, finite, and standalone vocabulary. We introduce a novel method that selects context-aware phrases from a collection of supporting documents. One of the most significant challenges for this paradigm shift is determining the training oracles, because a string of text can be segmented in various ways and each segment can be retrieved from numerous possible documents. To address this, we propose to initialize the training oracles using linguistic heuristics and, more importantly, bootstrap the oracles through iterative self-reinforcement. Extensive experiments show that our model not only outperforms standard language models on a variety of knowledge-intensive tasks but also demonstrates improved generation quality in open-ended text generation. For instance, compared to the standard language model counterpart, our model raises the accuracy from 23.47% to 36.27% on OpenbookQA, and improves the MAUVE score from 42.61% to 81.58% in open-ended text generation. Remarkably, our model also achieves the best performance and the lowest latency among several retrieval-augmented baselines. In conclusion, we assert that retrieval is more accurate generation and hope that our work will encourage further research on this new paradigm shift.

## 1 INTRODUCTION

*Memorization or generalization, that is the question.*

Standard language models (LMs) break down the text generation process into sequential token predictions (Mikolov et al., 2010; Brown et al., 2020; OpenAI, 2022). Each token is a word (or sub-word) selected from a fixed, finite, and standalone vocabulary. To make the generation more attributable and accelerate the inference speed, Lan et al. (2023) propose a method named CoG that retrieves phrases from similar contexts, where the term "phrase" refers to any contiguous text segments of variable lengths. It is worth noting that, similar to other retrieval-augmented generation frameworks (Li et al., 2022; Asai et al., 2023), CoG still employs a two-stage pipeline, specifically document retrieval followed by grounded phrase extraction. The final performance is constrained by the quality and quantity of the return from the first stage. In this paper, we propose a new paradigm that completely removes the dependence on document retrieval. To our best knowledge, our work is the first that performs text generation through direct phrase retrieval.

One core challenge of adopting this novel approach is the construction of the training oracles. That is a function mapping a string of text to an action sequence for creating training examples. For a given text, there exist numerous different ways to segment it into phrases, with each potential phrase being retrievable from a vast array of documents. To better align the generation process and the supporting documents, we introduce a two-fold approach: first, we leverage linguistics-motivated heuristics to initialize the training oracles. Second, we implement a bootstrapping mechanism through iterative self-reinforcement, gradually refining the oracles with each iteration.

Unlike Lan et al. (2023) which only evaluates the generation fluency in open-ended text generation, we carry out comprehensive and rigorous evaluation in a wide range of knowledge-intensive

---

∗Work done during an internship at Tencent AI Lab.
†Corresponding author.

tasks, *e.g.*, open-domain question answering. Our proposed model exhibits superior zero-shot performance, outperforming the baseline method. For example, on the OpenbookQA dataset, our model dramatically improves upon base LM, presenting an increase in accuracy from 23.47% to 36.27% (Table 1). Our model also demonstrates improved quality in open-ended text generation, as evidenced by the improvement of 38.97% in the MAUVE score (Table 4). Moreover, it shows even better performance when switching to an enlarged (Table 2) or domain-specific (Table 3) phrase table, without any further training. In addition, our model attains the fastest generation speed among retrieval-augmented baselines (Table 4). We believe that our study can inspire future research to build more efficient and accurate LMs that harness the power of retrieval-based approaches.

In summary, the contributions of this paper can be summarized as follows:

- We introduce a new approach for language modeling that focuses on directly selecting context-aware phrases from a set of supporting documents.
- We propose a novel method for decomposing text generation into sequential next-phrase retrieval by linguistics-driven heuristics and iterative self-reinforced bootstrapping.
- We validate the effectiveness of our models on various downstream tasks, including open-domain and domain-specific question answering, as well as open-ended text generation, highlighting substantial improvements over standard LMs and several retrieval-augmented baselines.

## 2 A UNIFIED VIEW OF GENERATION AND RETRIEVAL

Standard language models (LMs) factorize the generation probability of a sequence $\mathbf{x} = [x_1, x_2, \ldots, x_n]$ into a series of conditional probabilities $p(\mathbf{x}) = \prod_{i=1}^{n} p(x_i|\mathbf{x}_{<i})$. Hence, the generation is often performed by repeatedly predicting the next token based on the generated sequence thus far (*i.e.*, prefix). The next-token prediction probabilities are computed as

$$p(x_i|\mathbf{x}_{<i}) = \frac{\exp(E_p(\mathbf{x}_{<i}) \cdot E_c(x_i))}{\sum_{x' \in V} \exp(E_p(\mathbf{x}_{<i}) \cdot E_c(x'))}, \tag{1}$$

where $E_p(\mathbf{x}_{<i})$ is a vector representation of the prefix $\mathbf{x}_{<i}$, $E_c(x)$ denotes a vector representation of the token $x$, and $V$ stands for the token vocabulary. Through the above notations, we can see that the standard LMs can be viewed as a dual-encoder matching network connecting different prefixes and tokens. Typically, as shown in the left part of Figure 1, the source encoder $E_p$ is implemented by a multi-layer neural network (e.g., Transformers) while the target encoder $E_c$ is simply a token embedding layer. As seen, the design of the dual-encoder network is heavily unbalanced; The source side is much more complex than the target side.

Recently, a retrieval-augmented LM, CoG (Lan et al., 2023), has been proposed. In addition to token selection, CoG also allows for phrase retrieval (i.e., variable-length $n$-grams) from a collection of supporting documents. From our point of view, CoG augments the target side of conventional LMs. First, the candidate pool is enlarged to include phrases of variable lengths. Second, the target encoder not only considers the candidates themselves but also their contexts.

However, searching phrases from large-scale corpora is resource-intensive. Therefore, CoG adopts a two-stage search strategy: relevant documents are first retrieved to reduce the search space for phrase selection. To construct the training oracles, CoG uses a forward maximum matching algorithm to find the longest matching phrases from the retrieved documents. Despite promising results, CoG cannot guarantee to provide a globally optimal solution for phrase retrieval, and is highly dependent on the external tool for document retrieval. In contrast, we present a new paradigm, which we call CoG-2, that generates text directly through phrase retrieval.

## 3 THE PROPOSED METHOD — CoG-2

### 3.1 OVERVIEW

Our research aims to enhance the interpretability and factuality of language models (LMs) by transitioning from token generation to phrase retrieval. First, the semantics of phrases are enhanced by their surrounding contexts (Mikolov et al., 2013), leading to a more discriminative representation for inference. Second, each retrieved phrase can be traced back to its original document, enhancing the accountability of the output.

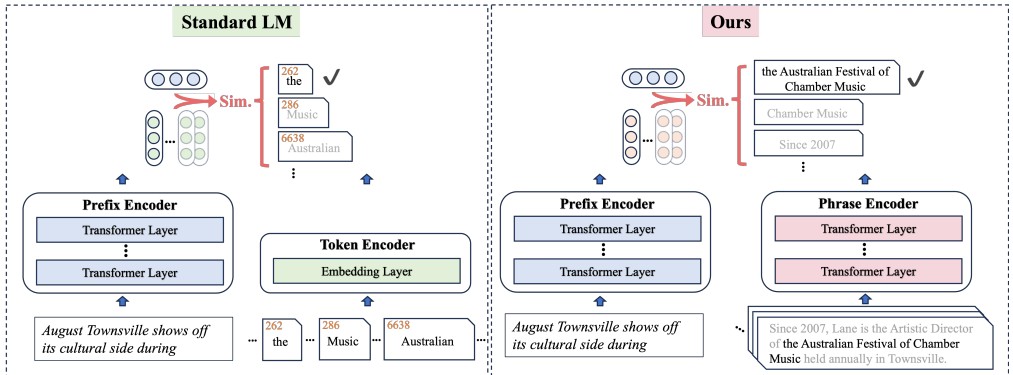

Figure 1: Comparison between our method and standard language models. Both can be viewed as dual-encoder matching networks connecting source prefixes and target continuations. On the target side, standard language models employ an immediate embedding layer for target tokens from a fixed, finite, and standalone vocabulary. In contrast, our methods uses an expressive phrase encoder for target phrase from an editable, extensible, and contextualized phrase table.

| Flag burning ... "flag burning" refers only to burning a flag as an act of protest. | | sends | | ... Each song has **a powerful message** designed to stop and make you think about your life ... | |
|---|---|---|---|---|---|
| Flag burning is a propaganda tool, such as burning Effigies of world leaders. | | sends | | the song ... sends **a powerful message** through its lyrics, telling listeners to 'keep going' and to fight for ... | |
| Flag burning ... situation escalated further after the parliamentary elections in ... | | ... for its "very bold move making tonight plant-based. It really **sends a powerful message**." Soon after, Critics' Choice and SAG ... | | | |
| Flag | burning | sends | a | powerful | message |

Figure 2: Four possible generation paths for the sentence "Flag burning sends a powerful message". Content highlighted in blue (red) are phrases retrieved from supporting documents (from the token vocabulary). Standard LMs can be viewed as only considering the generation path at the bottom.

To link a given prefix with a set of variable-length phrases, our model follows the dual-encoder structure as described in Section 2 but emphasizes a balanced design in contrast to standard LMs that heavily favor the source side (see Figure 1). Specifically, the source encoder $E_p(\cdot)$ is a multi-layer neural network (e.g., Transformer) as usual. The target encoder $E_c(\cdot)$ is also a multi-layer neural network to learn context-aware representation for phrases in supporting documents.

Similar to standard LMs, we employ dot product as the matching measure. During inference, we can use efficient maximum inner product search (MIPS) algorithms (Shrivastava & Li, 2014; Guo et al., 2016; Seo et al., 2019) to retrieve from a large pool of candidate phrases. The overall framework is depicted in Figure 1. The remaining question is how to train our models.

## 3.2 TRAINING ORACLES

We break down text generation into a series of next-phrase retrieval. Formally, each step takes the current prefix $p$ as its state, an oracle policy $\pi^*$ maps the state to an action $\pi^*(p) \rightarrow (f, s)$, where $f$ is a follow-up phrase and $s$ is a copy of the phrase $f$ in a supporting document.

As illustrated in Figure 2, to create such triplets $(p, f, s)$ from raw corpora presents two challenges. First, the boundary of the phrase $f$ is unclear given a continuation can be divided in various ways. Second, the source of each phrase $s$ is unclear because a phrase can appear numerous times across a vast number of documents. On the other hand, the variety of generation paths for a given text also indicates that training oracles are crucial for optimal and quick convergence of our models.

To tackle the above problems, we first present a set of linguistics-motivated heuristics to initialize the training oracles (Section 3.2.1), then describe how we allow the model to refine its generation paths in a self-reinforcement manner (Section 3.2.2).

### 3.2.1 Linguistics-motivated Heuristics

We start to design the training oracles through the following principles.

**Syntactic Structure.** Inspired by the syntactic structure of language and its implications on language generation (Chomsky, 1957; Dyer et al., 2016; Li et al., 2023b), we restrict the phrase to a contiguous sequence of words that corresponds to a constituent unit in a syntactic parse tree. This approach ensures that each phrase possesses a relatively complete and well-defined meaning, while avoiding arbitrary word combinations that could result in semantic ambiguity or nonsensical formations (Morgan & Newport, 1981).

**Distributional Sparsity.** The inclusion of high-frequency phrases significantly inflates the size of the candidate pool. This is due to our treatment of lexically identical phrases in different contexts as distinct entries in the pool. Consequently, a single high-frequency phrase could potentially introduce tens of thousands, or even millions, of entries. In our analysis of Wikipedia, we discovered that eliminating just the top 1% of high-frequency phrases could reduce the total number of entries by 50%. However, these high-frequency phrases, such as 'as well as', often lack specific meanings. Their inclusion may result in imbalanced training, which could adversely affect the model's overall performance. Regarding phrases with extremely low frequency, we consider them to be rare usages with limited practical use. Including them would notably increase the complexity of training. Therefore, we also choose to exclude them.

**Semantic Similarity.** Although a lexically identical copy of a phrase can be located in various places, it is crucial to account for polysemy (Cruse, 1986), as lexically identical phrases can exhibit different meanings depending on their contexts. Moreover, even when lexically identical phrases share similar meanings, subtle nuances can arise from different contexts, necessitating a thorough evaluation of semantic similarity when selecting the most appropriate matching (Min et al., 2019).

Specifically, we first run the Stanford Parser[1] to extract constituents from the training data. We then filter these constituents based on the following criteria: (1) remove trivial constituents with labels such as WHADJP, WHADVP; (2) exclude constituents that are too short ($< 2$ words) or too long ($> 10$ words); (3) discard constituents with excessively high or low Inverse Document Frequency (IDF) (Salton & Buckley, 1988) values. Notably, we apply a more lenient IDF threshold for longer constituents. Next, we group lexically identical phrases and compute the pairwise semantic similarities using BM25 (Robertson et al., 2009) and an off-the-shelf phrase encoder (Lee et al., 2021b). Consequently, we can identify the most suitable next phrase for each prefix based on the scores. For more detailed information, please refer to the Appendix A.

### 3.2.2 Iterative Self-reinforcement

The generation paths determined by the above heuristics are model-agnostic and could be noisy and sub-optimal (Welleck et al., 2019). To further improve performance, we allow the model to adjust its own generation paths based on the capabilities it has acquired. That is, transitioning from imitating the oracles to reinforcing its own preferences. In particular, we propose a bootstrapping algorithm to iteratively adjust the target phrases. For each prefix $p$, we first let the model retrieve the $k$-best phrases in the entire candidate pool using its current policy. Then, we choose the valid phrase with the highest semantic matching score from these $k$ phrases as the new target. If no such phrase is found, *i.e.*, none of the $k$-best phrases match the ground-truth continuation, we retain the previous target. The above process is repeated periodically. We present an example in Appendix B.

### 3.3 Training Objectives

We optimize our model using the InfoNCE loss (Oord et al., 2018; Karpukhin et al., 2020), for which a negative phrase set $\mathcal{N}(p)$ is introduced for each triplet $(p, f, s)$.

$$\mathcal{L}_p = \frac{\exp(E_p(p) \cdot E_c(s))}{\exp(E_p(p) \cdot E_c(s)) + \sum_{t \in N(p)} \exp(E_p(p) \cdot E_c(t))} \tag{2}$$

The construction of the negative phrase set $\mathcal{N}(p)$ is detailed below. To preserve the ability for token-level generation, we also train our model with the standard next-token prediction loss $\mathcal{L}_t$ (Lan et al., 2023). The training objective is formulated as $\mathcal{L}_p + \alpha \mathcal{L}_t$.

---

[1] https://stanfordnlp.github.io/stanza/

**Negative Sampling.** We incorporate two types of negative examples to improve the model's ability to differentiate phrases: (1) In-batch negatives: We regard all other candidate phrases in the same training batch as this type of negative example. These negatives help the model learn more discriminative representations on a large scale without incurring considerable costs. (2) Hard negatives: Recall that in Section 3.2.2, we periodically update the generation targets by retrieving top-$k$ candidate phrases for each prefix. Among these $k$ phrases, despite one may be chosen as the new generation target, the remaining phrases can serve as strong negatives because they are likely to confuse the model.

Note that the above negatives may contain false negatives, which are not chosen as targets but still make a valid follow-up. To minimize the risk, we remove all phrases that constitute a prefix of the groundtruth continuation.

### 3.4 MODELS

**Prefix Encoder.** We treat the prefix as a sequence of tokens with previously predicted phrases split into tokens. This token sequence is encoded using the standard Transformer architecture with causal attention (Vaswani et al., 2017; Radford et al., 2019). The prefix representation is obtained through a linear projection of the last-layer representation of the final token in the sequence.

**Phrase Encoder.** We employ a deep bidirectional Transformer (Vaswani et al., 2017; Devlin et al., 2019) to generate contextualized token representations of a supporting document. The representation of a phrase is obtained by concatenating the representations of its first and last tokens, followed by projecting the concatenated representation to the same dimension as the prefix representation. To preserve the ability to compose output using single tokens, we also add the token vocabulary to our phrase table. These standalone tokens can be considered as special phrases, and their representations are obtained through the standard embedding layer of the LM.

## 4 EXPERIMENT SETUP

### 4.1 IMPLEMENTATION DETAILS

We train our model on the training set of MiniPile[2](Kaddour, 2023), and use the English Wikipedia dump March 1, 2022[3] as supporting documents. Specifically, we split each Wikipedia article into multiple, disjoint text blocks of up to 128 words as documents, which results in 29,488,431 documents. The size of our phrase index is 137,101,097. We use GPT-2 (Radford et al., 2019) and DensePhrases[4] (Lee et al., 2021b) to initialize the prefix encoder and the phrase encoder, respectively. For efficiency, we solely fine-tune the prefix encoder. This avoid the computational burden of re-computing phrase embeddings associated with updating the phrase encoder. While revising the training oracles via self-reinforcement, we retrieve the top $k = 128$ phrases for each prefix.

### 4.2 INFERENCE DETAILS

During inference, we employ FAISS (Johnson et al., 2019), a library for vector similarity search and clustering, for efficient retrieval.

**Continuation Generation.** For text generation, we directly retrieve top-$k$ candidates from the entire phrase table (including both context-aware phrases and standalone tokens). We then apply a `softmax` function to the matching scores of these candidates, creating a next-phrase probability distribution (Shi et al., 2024), and use top-$p$ sampling (Holtzman et al., 2020) for selecting the next phrase. In all experiments, we set $k$ to 128 (see the analysis on $k$ in Table 7 in Appendix G) and $p$ to 0.95. To control the ratio of phrase retrieval, we filter out phrases with probabilities below a threshold. The threshold is set to $\phi = 0.4$ if not otherwise specified.

---

[2]`https://huggingface.co/datasets/JeanKaddour/minipile`

[3]`https://huggingface.co/datasets/wikipedia`

[4]`https://huggingface.co/princeton-nlp/densephrases-multi`

**Likelihood Estimation.** To calculate the likelihood of a given text, we approximate the likelihood by summing all possible generation paths. For instance, given the sentence *"The Moon rises"*, the following generation paths may exist: (1) *The→moon→rises*; (2) *The moon→rises*; (3) *The moon rises*. The probability of each path is the product of the probabilities of all phrases (tokens) along that path. For example, the probability of the path (2) is calculated by $p(rises|The\ moon) \cdot p(The\ moon)$. The probabilities of each step are obtained in the same way as we construct the next-phrase probability distribution for continuation generation. Note that the sum of all possible paths can be computed efficiently using dynamic programming with time complexity $O(n^2)$, where $n$ represents the number of tokens in the text.

## 4.3 BASELINES

We compare the proposed method with standard LM in the zero-shot setting, also drawing the following state-of-the-art retrieval-augmented methods as baselines:

**Base LM** is the standard token-level language model using the Transformer (Vaswani et al., 2017) architecture. We fine-tune the pre-trained GPT-2[5] (Radford et al., 2019).

$k$**NN-LM** (Khandelwal et al., 2020) is a retrieval-augmented LM that interpolates the next-token distribution of the base LM with a $k$-nearest neighbors ($k$NN) model.

**RETRO** (Borgeaud et al., 2022)[6] is a retrieval-augmented LM incorporated with a pre-trained document retriever, a document encoder and a cross-attention mechanism.

**CoG** (Lan et al., 2023)[7] is another retrieval-augmented LM that adopts a two-stage search pipeline. It first retrieves semantically-relevant documents, and then considers all $n$-grams within them as candidate phrases.

## 5 EXPERIMENTS

We verify the effectiveness of our methods on a set of knowledge-intensive tasks and open-ended text generation tasks without fine-tuning.

### 5.1 KNOWLEDGE-INTENSIVE TASKS

#### 5.1.1 DATASETS

We employ five knowledge-insensitive datasets, including three open-domain QA datasets: **OpenbookQA** (Mihaylov et al., 2018), **ARC-Challenge** (Clark et al., 2018), and **TruthfulQA** (Lin et al., 2022); and two domain-specific (medical) datasets: **MedMCQA** (Pal et al., 2022) and **MedUSMILE** (Jin et al., 2021). The details for these datasets can be found in Appendix C.

In line with prior research (Brown et al., 2020; Sanh et al., 2022), we adopt a *classification with options* methodology to quantify the model performance. This approach involves presenting the model with a range of options and calculating the likelihood of each option being the correct response. The option with the highest probability is selected as the model's prediction. We then report the **accuracy** of the model's predictions.

#### 5.1.2 RESULTS

We compare our methods with baselines in knowledge-intensive tasks across several settings.

**Main Results.** As shown in Table 1, our model consistently outperforms various baseline models across all datasets. Compared with base LM, our model improves the accuracy of the TruthfulQA and OpenBookQA datasets from 29.73% to 34.27% and 23.47% to 36.27%, respectively. When we eliminate the phrase retrieval from our model and only use standalone tokens (Ours w/o phrase), there is a considerable drop in performance, demonstrating the effectiveness of incorporating phrase

---

[5]https://huggingface.co/gpt2

[6]https://github.com/lucidrains/RETRO-pytorch

[7]https://github.com/gmftbyGMFTBY/Copyisallyouneed

| | TruthfulQA | OpenbookQA | ARC-Challenge | MedMCQA | Med-USMILE |
|---|---|---|---|---|---|
| Base LM (w/o FT) | 30.27 | 22.67 | 24.52 | 27.96 | 24.89 |
| Base LM | 29.73 | 23.47 | 23.92 | 28.33 | 24.19 |
| *k*NN-LM | 30.27 | 22.93 | 24.82 | 27.96 | 24.72 |
| RETRO | 27.53 | 26.13 | 22.21 | 25.68 | 25.33 |
| CoG | 34.11 | 35.47 | 27.24 | 29.07 | 25.07 |
| Ours | **34.27** | **36.27** | **28.27** | **29.44** | **25.69** |
| Ours(w/o phrase) | 28.63 | 23.73 | 22.51 | 27.42 | 24.80 |

Table 1: Experiments on knowledge-intensive tasks. Ours (w/o phrase): a variant of our model that restricts the model to only use standalone tokens without retrieving context-aware phrases.

| | TruthfulQA | OpenbookQA | ARC-Challenge | MedMCQA | Med-USMILE |
|---|---|---|---|---|---|
| Ours | 34.27 | 36.27 | **28.27** | 29.44 | 25.69 |
| w/ enlarged index | **39.59** | **37.07** | 27.14 | **31.63** | **27.87** |

Table 2: Results for our model with an enlarged phrase index.

retrieval in our methods. Note that the models presented in Table 1 are initialized from pre-trained LMs. To analyze the role of pre-trained models in our framework, we train all models from scratch with random initialization. The results are shown in Table 8 in Appendix G, our model outperforms the baselines across all datasets. For example, our model achieves a 12.8% absolute improvement on OpenbookQA over base LM, suggesting that our training framework is not heavily dependent on pre-trained models. To elucidate the role of phrase retrieval in knowledge-intensive tasks, we delve into a case study depicted in Appendix D.

**Enlarged Phrase Index.** Recall that we exclude phrases with excessively high or low IDF values (Section 3.2.1). This strategy not only stabilizes the training process but also improves training efficiency. However, the phrases initially filtered out can be repurposed to augment our phrase index in a training-free manner. This expanded phrase index, now three times larger than the original, underscores the scalability of our approach. As evidenced in Table 2, this expansion boosts our model's performance, such as a 5.32% increase in accuracy on TruthfulQA. This not only highlights our model's potential to generalize to unseen phrases and documents but also emphasizes its plug-and-play feature, capable of adapting to a larger phrase table without the need for re-training.

**Domain Adaption.** The *plug-and-play* property of the phrase index further motivates us to employ a domain-specific index for the QA tasks in the medical domain without any domain-specific training. To this end, we construct an index consisting of 3 million phrases by extracting phrases from a small text collection of the medical domain[8]. For comparison purpose, we also

| | MedMCQA | Med-USMILE |
|---|---|---|
| Base LM (FT) | 28.79 | 25.15 |
| General index | 29.44 | 25.69 |
| Medical index | **29.50** | **26.38** |
| w/o phrase | 27.42 | 24.80 |

Table 3: Results on medical datasets.

fine-tune the base LM on it for fair comparison. As illustrated in Table 3, despite the considerable reduction in index size compared to the original Wikipedia index (3 million vs 137 million), our model exhibits even better performance on two medical QA datasets. This result underscores our model's capability to enhance its performance in specific domains by leveraging a domain-specific, well-curated phrase index in a training-free manner.

## 5.2 OPEN-ENDED TEXT GENERATION

We conduct open-ended text generation experiments on the test set of MiniPile (Kaddour, 2023). For each document in the test set, we adopt the first 128 tokens as the prefix. The baselines and our model are required to generate text continuations of 128 tokens in length based on the same prefix.

---

[8]https://huggingface.co/datasets/gamino/wiki_medical_terms

| | MAUVE↑ | Coherence↓ | Diversity↑ | Latency↓ |
|---|---|---|---|---|
| Base LM (w/o FT) | 69.68 | 3.64 | 83.14 | 1.00x |
| Base LM | 42.61 | 3.56 | 78.72 | 1.00x |
| $k$NN-LM | 13.07 | 5.63 | **88.10** | 6.29x |
| RETRO | 62.39 | 4.82 | 80.96 | 1.51x |
| CoG | 52.27 | **2.08** | 55.04 | 4.40x |
| Ours | **81.58** | 3.25 | 76.26 | **1.29x** |

Table 4: Results for open-ended text generation.

| Model | Fluency | Coherence | Informativeness | Grammar |
|---|---|---|---|---|
| Base LM (w/o FT) | 2.91 | 2.33 | 2.35 | 3.00 |
| Base LM | 2.81 | 2.37 | 2.40 | 2.79 |
| Ours | **2.95** | **2.70** | **2.67** | **3.02** |

Table 5: Human evaluation results.

### 5.2.1 EVALUATION METRICS

Following previous works (Welleck et al., 2020; Su et al., 2022; Lan et al., 2023), we utilize three automatic evaluation metrics to measure the quality of the generated texts: (i) **MAUVE** (Pillutla et al., 2021) captures the overall usefulness of the generated text by estimating the average utility of the content; (ii) **Coherence** measures the logical consistency and flow of the generated text, ensuring that the output is well-structured and easy to understand; and (iii) **Diversity** evaluates the variety of generated content, promoting the generation of unique and creative text. We report MAUVE and diversity as percentages (%). The details for these metrics can be found in Appendix E. We also measure the average time cost for a model to decode a continuation consisting of 128 tokens given a prefix of 128 tokens, referred to as **latency**.

### 5.2.2 RESULTS

As shown in Table 4, our model attains the highest MAUVE score among all models, demonstrating the high quality of the generated text. Other retrieval-augmented methods underperform base LM in the MAUVE score due to text degeneration, which aligns with findings in previous work (Wang et al., 2023). Our model also shows a strong balance between coherence and diversity. The coherence score of our model is 3.25, which outperforms most baselines except for CoG. However, we find that CoG often generates lexically similar, meaningless sentences, which is reflected in its low diversity score of 55.04%. Meanwhile, our model's diversity score is 76.26%, which is slightly lower than some baseline models, but these models often generate incoherent sentences, as reflected in their lower coherence scores.

**Human Evaluation.** To gain further insights, we randomly sample 100 cases and evaluate the results of the base LM, the base LM without fine-tuning (w/o FT), and our model from four perspectives: fluency, coherence, informativeness, and grammar. Each aspect is scored on a Likert scale from 1 to 4 (1 represents *"bad"*, 2 stands for *"fair"*, 3 is considered *"good"*, and 4 signifies *"very good"*). We report the average scores in table 5. As we can see, our method outperforms the base LM in all four categories, especially in coherence and informativeness. This indicates that our model, based on phrase retrieval, is better at following the preceding context and providing more informative content. As for the lower scores of the base LM compared to the base LM (w/o FT), we find that they are largely due to formatting issues. Further analysis can be found in Appendix F.

**Generation Speed.** We now discuss the generation latency of different models. In Table 4, we report the relative latency, taking the base LM as the baseline. $k$NN-LM incurs the highest cost due to the need for interpolating the base LM's token distribution with another distribution computed using its datastore. The CoG model also exhibits a notable overhead as it involves extracting all $n$-grams from the retrieved documents, applying `softmax` over tokens and all $n$-grams, and sampling from the resulting probability distribution. The RETRO model, although faster than the previous two, still

requires time for applying the representations of retrieved text chunks in attention computation. Our method stands out with the highest generation speed, since it directly retrieves and utilizes phrases.

**Effect of Self-reinforcement.** Ablation studies on the effect of the Self-Reinforcement (SR) mechanism reveal significant insights into the performance of our model. In the case of knowledge-intensive tasks, we do not observe a significant impact of SR on our model's performance (refer to Table 9 in Appendix G). This suggests that our framework is inherently effective in handling such tasks, even without the aid of SR. However, the scenario differs for open-ended text gen-

|        | MAUVE↑ | Coh.↓ | Div.↑ |
|--------|--------|-------|-------|
| w/o SR | 7.86   | 4.14  | **81.14** |
| round1 | 64.49  | **3.23** | 70.15 |
| round2 | **81.58** | 3.25 | 76.26 |

Table 6: Ablation study on the effect of self-reinforcement.

eration. Table 6 shows that models trained with SR exhibit substantial improvements in the MAUVE scores across multiple rounds, which indicates the importance of SR in enhancing the quality of text generation. After the second round, we do not observe noticeable improvements with additional rounds of SR iteration, suggesting that the model converges to its optimal state.

## 6 RELATED WORK

Standard language models (LMs) (Radford et al., 2019; Brown et al., 2020) are trained to predict the next token given a text prefix. With a vast amount of training corpora and model parameters, these models show strong zero-shot performance on various downstream tasks, serving as a unified solution for natural language processing. However, scaling up the model parameters and training corpora can be very expensive and cannot be done in a timely manner.

To tackle the above issues, there has been an increasing body of work that enhances the parametric LM with a non-parametric component (Li et al., 2022). Guu et al. (2020); Lewis et al. (2020); Borgeaud et al. (2022); Izacard et al. (2022) ground the next token prediction on a set of relevant documents obtained using retrieval techniques (Robertson & Zaragoza, 2009; Karpukhin et al., 2020). Khandelwal et al. (2020); Yogatama et al. (2021); Zhong et al. (2022) augment the output probability distribution with non-parametric nearest neighborhood estimation. Also, the retrieve-then-generate paradigm has been extensively studied in specific downstream tasks, such as code generation (Hashimoto et al., 2018), question answering (Ye et al., 2023; Karpukhin et al., 2020; Lee et al., 2021a), open-domain dialogue systems (Weston et al., 2018; Wu et al., 2019; Cai et al., 2019a;b), and machine translation (Khandelwal et al., 2021; Cai et al., 2021), multimodal retrieval (Jin et al., 2023; Li et al., 2023a).

The work most closely related to ours is that of Min et al. (2022) and Lan et al. (2023). The former explores a similar idea in the area of masked language models to enhance natural language understanding. Lan et al. (2023), on the other hand, allows the copy of phrases from the grounding documents. However, their approach still relies on a two-stage pipeline, grounding the generation on a small set of retrieved documents only. While Lan et al. (2023) simply employs the longest common subsequence algorithm to find phrases that can be copied from the retrieved documents, we present heuristics-based and self-reinforced mechanisms to construct reliable training oracles. Also, Lan et al. (2023) only evaluates the performance on open-ended text generation tasks.

## 7 CONCLUSION

We presented CoG-2, a novel retrieval-based text generation approach using context-aware phrase retrieval. Our method addresses the primary challenge of constructing training oracles through heuristic-based initialization and iterative self-reinforcement. Experiments on knowledge-intensive tasks and open-ended text generation tasks show that the proposed method outperforms the standard LM and state-of-the-art retrieval-augmented methods. Moreover, our model exhibits superior performance with either an enlarged or a smaller, domain-specific index, and achieves the lowest generation latency compared to other retrieval-augmented baselines. This work contributes to the NLP research community by promoting a paradigm shift towards more accurate generation via retrieval. As we continue to explore and refine the paradigm, we invite readers to consider the limitations of our current work, as detailed in Appendix H, to fully appreciate the scope of future research.

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

## A  PHRASE TABLE PRUNING AND PHRASE MATCHING

It is noteworthy that syntactic parsing is a very well-studied task in NLP as well as its cross-domain and cross-language generalization. For example, the Universal Dependencies [9] project provides consistent grammatical annotation across over 100 languages. To our knowledge, the state-of-the-art parsing accuracies are pretty high for major languages such as English, Chinese, Italian, Japanese, Portuguese, etc. Nevertheless, we anticipate performance degradation for languages and domains when the parser accuracy is relatively low. For situations where a syntactic parser is unavailable, alternative methods may be utilized such as unsupervised syntactic parsing and unsupervised tokenization methods (*e.g.*, BPE, sentencepiece).

After extracting constituents from the training data and supporting documents, we filter these constituents based on the following criteria: (1) remove trivial spans with the following constituent labels: 'X', 'PRT', 'CC', 'DT', 'EX', 'FRAG', 'GW', 'HYPH', 'IN', 'INTJ', 'LS', 'LST', 'MD', 'NFP', 'NML', 'PDT', 'POS', 'PP', 'PRP', 'PRP$', 'PPZ', 'RB', 'RBR', 'RBS', 'RP', 'S', 'SYM', 'TO', 'WDT', 'WHADJP', 'WHADVP', 'WHNP', 'WHPP', 'WP', 'WP$', 'WRB', '#', '$', '""', '""', '-LRB-', '-RRB-', ',', '.', ':'; (2) exclude constituents that are too short ($< 2$ words) or too long ($> 10$ words); (3) discard constituents with excessively high or low Inverse Document Frequency (IDF) values. The (minimum, maximum) thresholds for constituents with different numbers of words are: 2: (10.50, 14.08), 3: (11.09, 14.08), 4: (11.77, 14.30), 5: (12.10, 14.30), 6: (12.32, 14.30), 7: (12.51, 14.59), 8: (12.59, 14.59), 9: (12.64, 14.59), 10: (12.69, 14.59). Next, we group lexically identical phrases and retrieve the top-10 candidates for each phrase using the BM25 algorithm (Robertson et al., 2009). We then calculate the semantic similarities between the original phrase and the retrieved candidate phrases using an off-the-shelf phrase encoder (Lee et al., 2021b). As a result, we can identify the most appropriate next phrase for each prefix based on the scores.

The entire preprocessing process, including syntactic parsing, phrase selection, and semantic matching, takes approximately 24 hours on 8 V100 GPUs. The overhead is small compared to the cost of training the model.

## B  EXAMPLE FOR ITERATIVE SELF-REINFORCEMENT

Suppose we have a prefix p = *"Go right for the top when you"*. The ground truth for this prefix is *"Go right for the top when you want to make things happen"*. The initial target phrase determined

---

[9]https://universaldependencies.org/

---

**Multiple-choice Question**

A 16-year-old girl is brought to the physician by her father because of concerns about her behavior during the past 2 years. She does not have friends and spends most of the time reading by herself. Her father says that she comes up with excuses to avoid family dinners and other social events. She states that she likes reading and feels more comfortable on her own. On mental status examination, her thought process is organized and logical. Her affect is flat. Which of the following is the most likely diagnosis?

[A] Schizoid personality disorder [B] Antisocial personality disorder [C] Schizophreniform disorder [D] Autism spectrum disorder

---

**Retrieved Phrases**

- Schizoid personality disorder (SPD) is characterized by a lack of interest in social relationships, a tendency towards a solitary lifestyle, secretiveness, emotional coldness, and apathy …
- Schizotypal personality disorder is characterized by a need for social isolation, anxiety in social situations, odd behavior and thinking, and often unconventional beliefs. People with this disorder feel extreme discomfort with maintaining close relationships with people, and therefore they often do not.

---

Figure 3: An illustrative example from Med-USMILE: The two highlighted phrases in red are retrieved in response to the posed question.

by the heuristics might be *"want"*. In the iterative self-reinforcement process, we would first let the model retrieve the $k$-best phrases for the prefix from the entire candidate pool. Supposing that the $k$-best phrases are [*"want"*, *"want to"*, *"want to make things happen"*, *"need"*, *"can"*], only *"want"*, *"want to"*, and *"want to make things happen"* are considered as valid ones. If the model's semantic matching score is highest for *"want to make things happen"*, we would update the target phrase for the prefix to this phrase. If none of the $k$-best phrases are valid, we will retain the previous target *"want"*.

## C   DETAILS OF TASK PHRASING AND SPECIFICATIONS

The statistics of the datasets we select are as follows:

**OpenbookQA** (Mihaylov et al., 2018) is a collection of 5,957 multiple-choice questions, each with four options, centered around elementary scientific knowledge. We utilize the test split, which comprises 500 questions.

**ARC-Challenge** (Clark et al., 2018) includes 7,787 authentic, grade-school level, multiple-choice science questions. These questions span a wide range of topics in science and history, among others. Our experiments focus on the test split of its Challenge Set, which contains 1,172 hard questions.

**TruthfulQA** (Lin et al., 2022) is a distinctive dataset emphasizing the truthfulness of answers. We employ the test split of the multiple-choice option, which includes 817 questions.

**MedMCQA** (Pal et al., 2022) is a comprehensive, high-quality dataset designed for biomedical question-answering. We use its validation split, which consists of 4,183 questions.

**Med-USMILE** (Jin et al., 2021) encompasses 12,723 multiple-choice questions, each with four options, originally sourced from the National Medical Board Examination in the USA. We utilize its test split, which includes 1,273 questions.

Given a question with several candidate answers, we concatenate the question with each candidate answer to form options, and then ask the model to select the most accurate one among all the options. We remove questions from these datasets where all candidate answers are single words to ensure the inclusion of phrases in the retrieval process.

## D   CASE STUDY

To elucidate the role of phrase retrieval in knowledge-intensive tasks, we delve into a case study depicted in Figure 3. As previously discussed in Section 4.2, our approach involves retrieving phrases for each token in an option, enabling us to estimate the probabilities of alternative generation paths beyond simply generating the token sequence. In this specific case from the Med-USMILE dataset, options are formed by concatenating the question with each candidate answer. We find that the phrases retrieved for the final token of the question include the answer, a proper noun requiring medical knowledge for understanding. This introduces a new generation path:

| $k$ | TruthfulQA | OpenbookQA | ARC-Challenge | MedMCQA | Med-USMILE | Avg. |
|------|------------|------------|---------------|---------|------------|------|
| 1 | 32.74 | **36.80** | 27.94 | **29.95** | 25.68 | 30.62 |
| 2 | 32.88 | 36.80 | 28.04 | 29.90 | 25.68 | 30.66 |
| 4 | 33.29 | 36.80 | 27.84 | 29.84 | 25.68 | 30.69 |
| 8 | 33.42 | 36.80 | 27.84 | 29.76 | 25.42 | 30.65 |
| 16 | 34.25 | 36.80 | 27.64 | 29.61 | 25.15 | 30.69 |
| 32 | 34.11 | 36.27 | 27.64 | 29.50 | 26.12 | 30.73 |
| 48 | **34.38** | 36.27 | 28.04 | 29.27 | **26.21** | **30.83** |
| 64 | 33.84 | 36.53 | **28.34** | 29.38 | 25.59 | 30.74 |
| 128 | 34.27 | 36.27 | 28.24 | 29.44 | 25.69 | 30.78 |
| 256 | 33.42 | 36.27 | 27.37 | 29.24 | 24.80 | 30.22 |
| 512 | 32.88 | 35.73 | 27.64 | 29.33 | 25.68 | 30.25 |
| 768 | 32.47 | 35.47 | 27.74 | 29.67 | 25.42 | 30.15 |
| 1024 | 32.47 | 35.47 | 27.54 | 29.61 | 24.89 | 30.00 |

Table 7: Ablation studies on the impact of $k$ on knowledge-intensive tasks.

|  | TruthfulQA | OpenbookQA | ARC-Challenge | MedMCQA | Med-USMILE |
|--|------------|------------|---------------|---------|------------|
| Base LM | 30.14 | 22.40 | 22.41 | 28.27 | 23.58 |
| $k$NN-LM | 30.14 | 22.40 | 23.32 | 27.99 | 23.14 |
| CoG | 32.88 | 34.13 | 25.13 | 29.16 | 25.15 |
| Ours | **33.29** | **35.20** | **27.04** | **30.24** | **26.21** |
| Ours (w/o phrase) | 28.22 | 21.87 | 23.02 | 27.99 | 24.89 |

Table 8: The results of models trained from scratch.

question $\rightarrow$ *Schizoid personality disorder*. We observe that the contexts of the retrieved phrases, such as "*Schizoid personality disorder (SPD) is characterized by a lack of interest in social relationships ...*", align closely with the context of the question, "*She does not have friends and spends most of the time reading by herself ...*". These contextually encoded phrases benefit answer selection, thereby showcasing the interpretability of our model. It also highlights the model's ability to leverage contextual information effectively, particularly in tasks that require specialized knowledge.

## E  DETAILS FOR AUTOMATIC EVALUATION METRICS

In this section, we provide a detailed introduction to MAUVE, as well as the concepts of coherence and diversity.

**MAUVE** (Pillutla et al., 2021) measures how closely the token distribution in generated text matches that in human-written text across the entire test set.

**Coherence** (Su & Collier, 2022; Su et al., 2022) measures the semantic coherence between the prompt $x$ and the generated text $\hat{x}$ by calculating the average log-likelihood as: $\text{coherence}(\hat{x}; x) = \frac{1}{|\hat{x}|} \sum_{i=1}^{|\hat{x}|} \log p_M(\hat{x}_i | [x : \hat{x}_{<i}])$, where $[:]$ is the concatenation operation and $M$ is a pre-trained LM. We follow prior work and set $M$ as the OPT-2.7B model (Zhang et al., 2022). In our implementation, we introduce a slight modification by taking the negative of the average log-likelihood. This adjustment transforms the typically negative log-likelihood into a positive value, facilitating a more intuitive interpretation of the results.

**Diversity** (Welleck et al., 2020; Su et al., 2022; Lan et al., 2023) measures the repetition in generated text at different $n$-gram levels by computing the proportion of unique $n$-grams to total $n$-grams in the generated text. It is defined as: $\text{diversity} = \prod_{n=2}^{4}(1.0 - \frac{\text{rep-n}}{100})$, where $\text{rep-n} = 100 \times (1.0 - \frac{|\text{unique n-grams}(\hat{x})|}{|\text{total n-grams}(\hat{x})|})$, and $\hat{x}$ is the text generated by the model.

|          | TruthfulQA | OpenbookQA | ARC-Challenge | MedMCQA | Med-USMILE |
|----------|------------|------------|---------------|---------|------------|
| w/o SR   | 34.11      | **37.07**  | 27.14         | **30.32** | **25.85** |
| round1   | 33.97      | 36.80      | 27.34         | 29.84   | 25.77      |
| round2   | **34.27**  | 36.27      | **28.24**     | 29.44   | 25.69      |

Table 9: Ablation studies on the effect of self-reinforcement.

## F   DETAILED HUMAN EVALUATION

Following the results in Table 5, upon manual analysis, we find that the outputs from our method often have a tighter connection with the preceding text (coherence) and exhibit stronger knowledge characteristics (informativeness). For instance, they often include specialized terms, which are not observed in the outputs from the base LM. As for the lower scores of the base LM compared to the base LM (w/o FT), one major reason is that the fine-tuned model frequently outputs "References: xxx" and "External Links: xxx". This is related to the characteristics of the Wikipedia dataset, where each article typically ends with references and external links. To further investigate this, we retest the Base LM, excluding all outputs containing "References" and "External Links" (which accounts for 26.77% of the cases). The resulting MAUVE score is 60.23, slightly lower than the 69.68 of the base LM (w/o FT), but substantially higher than the previous score of 42.61. This also suggests that one possible reason for the significant drop in the MAUVE score after (*c.f.* Table 4) fine-tuning the base model is due to these extraneous outputs.

## G   ABLATION STUDIES

**The impact of $k$.**   As shown in Table 7, $k$ does not have a significant impact on the performance of our model on knowledge-intensive tasks. Since we retrieve the top 128 phrases during the self-reinforcement process, we set $k = 128$ throughout all experiments.

**The impact of pre-trained models.**   The results are given in Table 8. Our model outperforms the base LM across all datasets, achieving a 12.8% absolute improvement on OpenbookQA. This suggests that our training framework is not heavily dependent on pre-trained models.

**The impact of phrase retrieval threshold $\phi$.**   The phrase retrieval threshold, which filters out phrases with probabilities below it, influences the proportion of tokens (token rate for short) used in the text generation process. This section explores the intriguing relationship between token rate and the quality of the generated text.

A lower token rate boosts the model's inference efficiency, as phrases typically contain multiple tokens, thereby reducing the number of decoding steps. However, this efficiency can sometimes compromise the quality of the text. This degradation occurs due to the inherent uncertainty during generation, particularly when all candidate probabilities are low or when an inappropriate candidate is sampled. In such cases, the selection of an incorrect phrase, given its length, can significantly disrupt the generation process, a phenomenon known as exposure bias (Ranzato et al., 2016; Zhang et al., 2019). Conversely, the impact of choosing a sub-optimal token is less severe. Therefore, up to a certain point (approximately 0.89), increasing the token rate can stabilize the generation process, as shown in Figure 4. Beyond this point, the model's performance peaks.

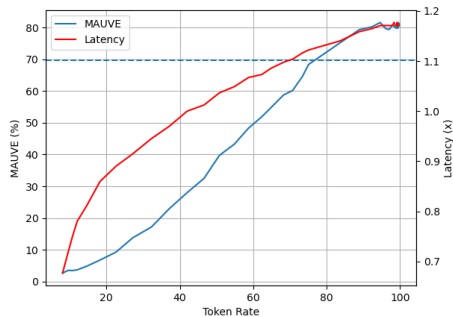

Figure 4: The MAUVE score and latency of our model with different token rates.

Furthermore, even without phrases (*i.e.*, the token rate is 1), our model can generate high-quality text, suggesting that our method also enhances token prediction learning. However, it's important

to note that while our model achieves a high MAUVE score based solely on token prediction, the factuality of the generated text is lower than when phrase retrieval is integrated. This highlights the need for more innovative metrics to precisely measure the quality of generated text.

**The impact of self-reinforcement on knowledge-intensive tasks.**    Based on the results presented in Table 9, it can be concluded that the utilization of SR does not significantly affect the performance of our model on knowledge-intensive tasks. This implies that our framework is inherently capable of effectively addressing such tasks, even in the absence of SR.

## H    LIMITATIONS AND FUTURE WORK

While our proposed framework has shown promising results in efficiently generating accurate and coherent text, it is important to acknowledge the limitations of its current form. First of all, the presented result is just a proof-of-concept of the new paradigm. Future work should focus on (1) Scalability. In our current experiments, we train our models and build the phrase index on the English Wikipedia corpus. When scaling up to larger corpora, we may encounter computational challenges due to the significantly increased amount of possible phrases. To make our approach scalable, some potential solutions include clustering, dimensionality reduction, and fast vector search algorithms with sub-linear time complexity. (2) Alignment. Recent studies have shown that alignment (i.e., fine-tuning language models to following human instructions) is important to make language models universally useful. Thus, incorporating alignment techniques into our approach is an important future research direction.

