# OpenReview forum: "Retrieval is Accurate Generation"
_ICLR.cc/2024/Conference — ICLR 2024 poster_

### Official Review · Reviewer_6EhH · 2023-10-24

**Soundness:** 3 good
**Presentation:** 3 good
**Contribution:** 3 good
**Rating:** 6
**Confidence:** 4

**Summary:**

The paper explores text generation, specifically by creating a larger vocabulary which additionally consists of phrases extracted from a large corpus via certain rules, and whose representations are formed by a transformer model encoding the wider context they appear in.
When decoding the model is then able to either generate standard tokens, or longer phrases.
Results are presented on open generation and question answer generation tasks, showing that the proposed model is able to perform more accurately than 3 other recent retrieval based baselines.

**Strengths:**

* Solid set of empirical results are presented showing the model performs well on the tested tasks.
* Comparisons against recent baselines are given.
* Additional phrase representations (produced by the same model used in training) are able to be added at inference time, with the decode model able to operate with these extra, new phrases.
* The method removes the inference-time dependence on document retrieval that other retrieval augmented generation papers have. It does move this phrase creation process (along with the embedding of these) to training time.

**Weaknesses:**

* The claims about inference speed are
* Can examples or further clarification be given for the 3.1 sentence "enhancing the accountability of the output"? This isn't clear, at least to me.
* There are a lot of heuristics in extracting the phrases. This may not be easy to repeat, or result in the same level of gains on other datasets or related problems.
* The decoding method seems very custom. Forcing a limited use of phrases, and blending top-k and top-p sampling. What happens if you just arg-max decode from the resulting model? Does it emit phrases way too often?
* Distributional sparsity  -- this section is not very clear.
* Is the likelihood estimation of summing paths well motivated? I'm not sure this is principled, but open to this being further justified.
* "For efficiency issues" in 4.1, does this mean for stability? Keeping the embeddings (of tokens and phrases) means the problem is stable I presume. I think this needs more explanation however.
* The numbers in table 1 are not described.

**Questions:**

* Was the vanilla LM trained with the phrase extended vocabulary? Or was this just using the gpt2 tokenisation alone? What happens if you  do this, rather than blending the contrastive phrase loss with the common CE loss?

---

> ### Author Response · Authors · 2023-11-20
>
> Thank you for your insightful review. We are delighted to learn that you believe our empirical results are “solid” and recognize several merits of our proposed methods as strengths.
>
> **W1:** *The claims about inference speed are*
>
> **A1**: It seems the sentence is incomplete. Could you please clarify your question? We will answer your question accordingly.
>
> **W2**: *Can examples or further clarification be given for the 3.1 sentence "enhancing the accountability of the output"? This isn't clear, at least to me.*
>
> **A2**: Thanks for your insightful question. We are happy to clarify this statement. When we say *"enhancing the accountability of the output"*, we're referring to the traceability of the phrases that our model generates. In traditional language models, the generated tokens are produced based on the model's internal representations, which can be difficult to interpret or trace back to the training data. However, in our approach, each phrase that is retrieved during the generation process can be directly traced back to its original document. This means that for each output of our model, we can provide a clear lineage or 'account' of where that output came from in terms of the supporting documents. This traceability enhances the accountability of the model, as it allows us to better understand and explain the model's outputs.
>
> - Example:
> Let's say we have a language model trained on a large corpus of news articles. When generating a news headline, traditional language models might produce a phrase like *"Breaking: New Study Shows Link Between Coffee and Cancer."* However, it can be challenging to determine where exactly this information came from within the training data. In our approach, when generating the same headline, our model retrieves phrases directly from the supporting documents. So, instead of just providing the headline, our model can trace back and indicate that the phrase *"New Study Shows Link Between Coffee and Cancer"* was retrieved from a specific news article published by a reputable scientific journal. This traceability enhances the accountability of the output, as it allows us to understand the source of the information and verify its credibility.
>
> **W3**: *There are a lot of heuristics in extracting the phrases. This may not be easy to repeat, or result in the same level of gains on other datasets or related problems.*
>
> **A3**: Thanks for your insightful question. We would like to clarify that the heuristics for phrase extraction are based on general linguistics principles like syntactic structure, semantic similarity, and distributional sparsity. In fact, our experiments are conducted on general corpora for language model pretraining (*i.e., MiniPile and Wikipedia*), without specific tailoring to any particular domain or problem. However, we acknowledge that some hyper-parameters may require tuning when scaling up the training data or altering the data mixture.
>
> **W4**: *The decoding method seems very custom. Forcing a limited use of phrases, and blending top-k and top-p sampling. What happens if you just arg-max decode from the resulting model? Does it emit phrases way too often?*
>
> **A4**: If we simply use argmax decoding, the model's output will contain a lot of repetitions, leading to text degeneration, a well-known issue for traditional language models as well. That's why we employ top-k and top-p sampling in our experiments. For your question on the emission rate of phrases, we conducted an experiment on the dev set of MiniPile. Specifically, we asked the model to predict the next token/phrase token-by-token. We found that **59%** of the time, the top-1 prediction is a phrase.
>
> **W5**: *Distributional sparsity -- this section is not very clear.*
>
> **A5**: Thanks for your constructive feedback. In our approach, we treat lexically identical phrases in different contexts as separate entries in the phrase pool. This means that each high-frequency phrase could potentially introduce tens of thousands, or even millions, of entries. During our analysis of Wikipedia, we observed that approximately 1% of the phrases have a frequency far exceeding that of the others. By removing just these **top 1%** high-frequency phrases, we can reduce the total number of entries by **50%**. Furthermore, we find that these top 1% phrases, such as *"as well as*", "it is," *"there are*", and others, often lack specific meanings and are context-independent. Therefore, we decided not to introduce these high-frequency phrases into our phrase pool, as they do not contribute significantly to the model's understanding of specific contexts and would lead to imbalanced training that may potentially affect the overall performance of the model.

---

> > ### Author Response · Authors · 2023-11-20
> >
> > **W6**: *Is the likelihood estimation of summing paths well motivated? I'm not sure this is principled, but open to this being further justified.*
> >
> > **A6**: The choice of “sum” is based on the intuition that each path represents a valid way of generating the sentence, and summing the scores allows us to account for all these possibilities. In our earlier experiments, we also tried to use the maximum score among all possible path (*i.e.,* “max”) and empirically found that the performances of “sum” and “max” are quite similar.
> >
> > **W7**: *"For efficiency issues" in 4.1, does this mean for stability? Keeping the embeddings (of tokens and phrases) means the problem is stable I presume. I think this needs more explanation however.*
> >
> > **A7**: Thank for your constructive question. *"for efficiency issues"* in section 4.1, we are primarily considering the computational cost. If we train the phrase encoder as well, we need to encode the supporting documents in each training batch on-the-fly to keep the phrase embeddings up-to-date. This would significantly increase the computational overhead during training. On the other hand, if we use a pretrained phrase encoder and do not update it, we can pre-encode all candidate phrases, which is much more efficient.
> >
> > **W8**: *The numbers in table 1 are not described.*
> >
> > **A8**: As mentioned in Section 5.1.1, we adopted a classification with options methodology to quantify the model performance and the numbers reported in Table 1 represent accuracy scores. We will add the details to the caption of Table 1 in the next version of this paper.
> >
> > **Q1:** *Was the vanilla LM trained with the phrase extended vocabulary? Or was this just using the gpt2 tokenisation alone? What happens if you do this, rather than blending the contrastive phrase loss with the common CE loss?*
> >
> > **A9**: Thanks for your insightful question. The vanilla language model was trained just using the GPT-2 tokenization alone, without expanding the vocabulary. Incorporating phrases into the vocabulary would result in a vocabulary size of 50k (original size) + 25369k (number of lexically unique phrases). Training a language model with such a big vocabulary incurs significant computational costs. On the other hand, we hypothesize that its performance may not be good. The reason is that most phrase embeddings cannot be well-trained due to data sparsity. Indeed, none of existing popular language models uses such a big vocabulary.
> >
> > Lastly, we would like to highlight the conceptual advantages of the proposed approach over simply increasing the vocabulary size. First, incorporating phrases into the vocabulary would assign a fixed representation for each phrase. In contrast, our phrase encoder approach retains more contextual information, allowing lexically identical phrases to have different embeddings depending on their contexts. This enables a more nuanced understanding of language. Second, the phrase encoder can be leveraged to obtain representations for phrases not encountered during training. This allows for better adaptation to different domains. Conversely, using a extended vocabulary with phrases would not easily accommodate the introduction of unseen phrases.

---

### Official Review · Reviewer_uQM4 · 2023-10-31

**Soundness:** 4 excellent
**Presentation:** 3 good
**Contribution:** 4 excellent
**Rating:** 8
**Confidence:** 4

**Summary:**

The paper introduces a novel method for language modeling that instead of generating tokens, retrieves phrases from a phrase-based index. This differs a lot from standard language models, which generate text by selecting tokens from a fixed, finite, and standalone vocabulary. Furthermore, this new approach leverages a more balanced encoding architecture for both the input and target tokens, as opposed to a single token embedding layer on the target side employed in standard language models. Moreover, their paradigm is the first that performs text generation through direct phrase retrieval, steering away from common 2-staged pipeline approaches, and thus removing the dependence on document retrieval and achieving lower latencies.
The authors shed light on how to determine the training oracles that allow this kind of training, and they propose to initialize them using linguistic heuristics. Also, in order to allow the model to adjust its own generation paths based on the capabilities it has acquired, they also bootstrap the oracles through iterative self-reinforcement, which gradually refines the oracles with each iteration by transitioning from imitating the oracles to reinforcing its own preferences.
In this new paradigm, text generation is achieved by copying retrieved phrases corresponding to constituent units in a syntactic parse tree, but the model still has the ability to generate individual tokens.
The effectiveness of their models is validated on various downstream tasks, including open-domain and domain-specific question answering, as well as open-ended text generation, attaining substantial improvements over standard LMs and several retrieval-augmented baselines. Transitioning to phrase retrieval improves interpretability and factuality on text generation tasks, as the semantics of phrases are enhanced by their surrounding contexts, and each retrieved phrase can be traced back to its original document. Finally, enlarging the phrase index during inference, and the plug-and-play feature of the index are shown to be effective and efficient methods for boosting the model's performance and adapting to out-of-domain distributions respectively, without any further training.

**Strengths:**

- A novel approach for retrieval augmented generation
- Holistic evaluation not only by measuring the fluency in open-ended text generation but also by carrying out comprehensive evaluation in a wide range of knowledge-intensive tasks, such as open-domain question answering.
- Plug-and-play feature of the phrase index, as a way of adapting to out-of-domain distributions (such as the Medical domain) by simply changing/extending the phrase index with a domain-specific index without any further training.
- Paper is generally well written and easy to follow
- Good explanation of how standard LLMs can be viewed as dual-encoding matching networks connecting different prefixes and tokens, and shedding light into the architecture imbalances between the prefix and the target encoders.

**Weaknesses:**

Weaknesses:
- More implementation details regarding the size of the phrase index, etc would be good to have in the paper.
- The work might also benefit from some discussion regarding scalability of the phrase index

Minor suggestions:

- As Figure 1 is the main overview of the approach proposed in the paper, a more detailed footnote would be appreciated.
- Section 6 "Results" wouldn't be better under subsection 5.2.2, as it reflects on results from the Open-Ended Text Generation experiments.
- Typos: Section 2, line 2. "The" after "Hence, " should be in lower-case.

Missing references:
- Minjoon Seo's work on phrase index QA: https://arxiv.org/pdf/1804.07726.pdf, https://arxiv.org/pdf/1906.05807.pdf


Overall I think this is an interesting method and the authors perform extensive experimentation to empirically justify their approach

**Questions:**

- Could you discuss any potential limitations or failure cases of the model, providing insights into scenarios where the proposed approach might not perform as effectively?

---

> ### Author Response · Authors · 2023-11-20
>
> Thanks for your recognition of our work. We very much appreciate your detailed review describing our approach as “novel” (”differs a lot from standard language models”), our evaluation as “holistic” and emphasizing the interpretability and “plug-and-play” features of our method. In the following, we would like to answer your concerns and questions one-by-one.
>
> **W1:** *More implementation details regarding the size of the phrase index, etc would be good to have in the paper.*
>
> **A1**: As mentioned in Section 5.1.1 (Domain Adaption), the size of our phrase index is **137,101,097**. We will make it clearer in the next version of this paper.
>
> **W2**: *The work might also benefit from some discussion regarding scalability of the phrase index*
>
> **A2**: Thank you very much for bringing up this insightful question! We consider the current results as a proof-of-concept for the proposed new paradigm. When scaling up to a larger phrase index, we certainly will face more computational challenges. To ensure scalability, we believe potential solutions are clustering, dimensionality reduction, quantization, and fast vector search algorithms with sub-linear time complexity, etc. We leave the implementation and exploration of these solutions as future work and will include this discussion in the next version of this paper.
>
> **Q1:** *Could you discuss any potential limitations or failure cases of the model, providing insights into scenarios where the proposed approach might not perform as effectively?*
>
> **A3**: One potential limitation is that our model relies heavily on the quality and coverage of the phrase index. If the phrases in the index are not diverse enough or do not cover certain fields, the model may struggle to generate accurate and coherent outputs for those topics. Therefore, in terms of failure cases, one scenario could be when the target text contains a lot of novel phrases or concepts that are not present in the phrase index. Another potential failure case could arise from the way we segment Wikipedia into documents of a maximum length of 128. This segmentation could result in some documents lacking sufficient context or information to effectively support the phrases within them. Our model relies on these supporting documents to obtain the phrase representations. If these documents are not clear or detailed enough, it could impact the model's performance. We are actively working on addressing these issues and improving our model's performance in these challenging scenarios.
>
> **Minor suggestions:**
>
> **M1**: *As Figure 1 is the main overview of the approach proposed in the paper, a more detailed footnote would be appreciated.*
>
> **A4**: Sure. we will use the following caption in the next version of this paper: “Comparison between our method and standard language models. Standard language models can be viewed as a dual-encoder matching network that connects different prefixes and tokens. In this architecture, the source encoder is implemented by a multi-layer neural network, while the target encoder is simply a token embedding layer. In contrast, our method adopts a balanced architecture, incorporating a phrase encoder to encode candidate phrases from supporting documents. As a result, we perform text generation through phrase retrieval, leveraging the contextual information captured by the phrase encoder.”
>
> **M2**: *Section 6 "Results" wouldn't be better under subsection 5.2.2, as it reflects on results from the Open-Ended Text Generation experiments.*
>
> **A5**: Thanks for pointing it out. It should be Section 5.2.2.
>
> **M3**: *Typos: Section 2, line 2. "The" after "Hence, " should be in lower-case.*
>
> **A6**: Thanks for pointing out the typos. We will fix them in the next version of this paper.
>
> **M4**: *Missing references: Minjoon Seo's work on phrase index* QA: https://arxiv.org/pdf/1804.07726.pdf, https://arxiv.org/pdf/1906.05807.pdf
>
> **A7**: We will include them in the next version of this paper.

---

### Official Review · Reviewer_iZXf · 2023-11-05

**Soundness:** 3 good
**Presentation:** 3 good
**Contribution:** 3 good
**Rating:** 8
**Confidence:** 4

**Summary:**

This paper combines retrieval with test generation and introduces an approach at retrieves context-aware phrases from a database of documents for generation. They use a set of linguistics heuristics combined with a bootstrapping method to extract phrases. The authors have done studies to show the effectiveness of their method and perform ablation study on the effect of different elements. They also study the inference speed of their approach and compare it with other approaches.

**Strengths:**

- The paper is well written and easy to follow.
- Their approach on text generation and selecting phrases is novel and introduces an interesting approach to text generation.
- The authors study the effectiveness of their approach well and provide comparisons with other approaches.
- Their zero-shot results on knowledge intensive tasks is convincing of the effectiveness of their approach.

**Weaknesses:**

- Lack of any human evaluations: Although there are automatic metrics for text generation, there still a need to have humans judge the generation.
- The paper does not provide deep insights into the observed results. For example, Section 6, Main Results, related to Table 4, it is not clear why the MAUVE score has such a huge jump for their method, or why finetuning the base model drops this score by a lot.

**Questions:**

- What corpus is used to finetune the base model?

---

> ### Author Response · Authors · 2023-11-20
>
> Thanks for your positive comments and describing our proposed approach as “novel” and “interesting”, and the experiment results on knowledge-intensive tasks “convincing”.  We provide our response to address your questions as below.
>
> **W1:** *Lack of any human evaluations: Although there are automatic metrics for text generation, there still a need to have humans judge the generation.*
>
> **A1:**  Thank you for your question. To address your concern, we conducted a human evaluation study on a random sample of 100 cases. We evaluate the results of the base LM, the base LM without fine-tuning (w/o FT), and our model from four perspectives: fluency, coherence, informativeness, and grammar. Each aspect is scored on a Likert scale from 1 to 4 ( 1 represents “*bad*”, 2 stands for “*Fair*”, 3 is considered “*good*”, and 4 signifies “*very good*”), and then we calculate the average scores. The results are as follows:
>
> | Model | Fluency | Coherence | Informativeness | Grammar |
> | --- | --- | --- | --- | --- |
> | Base LM (w/o FT) | 2.91 | 2.33 | 2.35 | 3.00 |
> | Base LM | 2.81 | 2.37 | 2.40 | 2.79 |
> | Ours | 2.95 | 2.70 | 2.67 | 3.02 |
>
> As we can see, our method outperforms the base LM in all four categories, especially in coherence and informativeness. Upon analysis, we find that the outputs from our method often have a tighter connection with the preceding text (coherence) and exhibit stronger knowledge characteristics (informativeness). For instance, they often include specialized terms, which are not observed in the outputs from the base LM.
>
> As for the lower scores of the base LM compared to the base LM (w/o FT), the issue lies in the fact that the fine-tuned model frequently outputs "References: xxx" and "External Links: xxx". This is related to the characteristics of the Wikipedia dataset, where each article typically ends with references and external links. These elements do not contribute positively to the perceived fluency and grammar.
>
> We will add the above discussion to the next version of this paper.
>
> **W2**: *The paper does not provide deep insights into the observed results. For example, Section 6, Main Results, related to Table 4, it is not clear why the MAUVE score has such a huge jump for their method, or why finetuning the base model drops this score by a lot.*
>
> **A2:** Following the above manual analysis (please refer to **A1**), we find that the main difference between the performance of the Base LM and the Base LM (w/o FT) is that the former often outputs "References: xxx" and "External Links: xxx".
>
> To further investigate this, we retest the Base LM on the test set of MiniPile, excluding all outputs containing "References" and "External Links" (which accounts for **26.77%** of the cases). The resulting MAUVE score is **60.23**, slightly lower than the **69.68** of the base LM (w/o FT), but substantially higher than the previous score of **42.61**. This suggests that the main reason for the significant drop in the MAUVE score after fine-tuning the base model is due to these extraneous outputs.
>
> As for the improvement in MAUVE for our method, this can be explained based on the notable improvements in terms of coherence and informativeness. This indicates that our model, based on phrase retrieval, is better at capturing the context and provides more informative content.
>
> We will add the above discussion to the next version of this paper.
>
> **Q1:** *What corpus is used to finetune the base model?*
>
> **A3**: We use the MiniPile and Wikipedia datasets to fine-tune the base model. Note that all baseline methods are trained using the same datasets.

---

### Official Review · Reviewer_Uubi · 2023-11-06

**Soundness:** 3 good
**Presentation:** 2 fair
**Contribution:** 3 good
**Rating:** 6
**Confidence:** 3

**Summary:**

Retrieval augmented generation models are very powerful  in making generation more attributable and trustworthy. The proposed approach in the paper belongs to this family. It is inspired from CoG (Lan et al.)  that retrieves phrases from similar contexts, however, unlike CoG, it doesn’t employ a two-stage pipeline, specifically document retrieval followed by grounded phrase extraction. The proposed approach removes the dependence on document retrieval.

Interestingly, the authors propose to use linguistics-motivated heuristics to initialize the training oracle phrases, followed by a bootstrapping mechanism through self-reinforcement to refine the oracle with each iteration. This linguistically inspired approach could be very useful in providing meaningful attributions to their sources.

The experiments on Open-book qa and open ended generation show consistent improvements over competitive baselines.

**Strengths:**

The proposed approach seems very interesting and will be useful to the generation community. As I mentioned earlier, the linguistically inspired approach could be very useful in providing meaningful attributions to their sources.

Strong results on a variety of benchmarks from Open book qa and open ended generation tasks.

==

Most of my concerns were adequately addressed in the authors rebuttal. Please include these details in the camera ready version if accepted. I have update my reviews accordingly.

**Weaknesses:**

The authors proposed a very interesting approach but I felt a lot of important details are missing. Please see my questions/comments below. It is unclear whether or not the code will be released from this work.

Another weakness of the work I believe is that this approach will not be robust to languages or domains where our syntactic parsing capabilities are limited.

**Questions:**

“each phrase possesses a relatively complete and well-defined meaning” -> Will this approach be not generalizable to languages and domains where the availability of syntactic parsers is limited? Also is it feasible to annotate the whole training set?

“Incorporating high-frequency phrases can significantly increase the total number of phrases, leading to an extremely large candidate pool” -> Won’t the low-frequency phrases significantly increase the size of the candidate pool?

Second paragraph under “Semantic similarity”: I felt lots of details were missing here to better understand the quality of phrases, and the feasibility of the proposed approach. The Appendix A do not provide all necessary details. Is this done on the pretraining corpus? What trivial constituents were dropped out and why (some examples would help)?

Sec 3.2.2: I found the explanation a bit confusing. Could you add an algorithm and/or an example demonstrating the algorithm?

“If no such phrase is found, we retain the previous target.” -> When would this occur? When the candidate pool is empty?

“we also add the token vocabulary to our phrase table” -> Are they subword units? What is the vocabulary?

“We train our model on the training set of MiniPile2 (Kaddour, 2023)” -> What is this dataset? Is it a pretraining set of finetuning set? How are they used during training? This dataset is discussed again in 5.2.

“Note that the sum of all possible paths can be computed efficiently using dynamic programming with time complexity O(n 2 ), where n represents the number of tokens in the text” -> Will this be limiting for long form outputs?

Table 1: Are the numbers for baselines taken from the respective papers or are they reproduced by the authors?

Sec 6: Results: This should be Sec 5.2.2?

---

> ### Author Response · Authors · 2023-11-20
>
> Thanks for your encouraging feedback and describing our work as “very interesting”, “very useful”, and “strong results”.  Below, we would like to address your concerns point by point.
>
> **W1**: *implementation details and code release*
>
> **A1**:  We will add more details to the paper and release our code upon acceptance.
>
> **W2**: *robustness to languages or domains where our syntactic parsing capabilities are limited.*
>
> **A2**: Thanks for this insightful question. Indeed, the initialization of our approach relies on syntactic parsing. We anticipate performance degradation for languages and domains when the parser accuracy is relatively low. However, we would like to mention that syntactic parsing is a very well-studied task in NLP as well as its cross-domain and cross-language generalization. For example, the Universal Dependencies (**https://universaldependencies.org/**) project provides consistent grammatical annotation across over 100 languages. To our knowledge, the state-of-the-art parsing accuracies are pretty high for major languages such as English, Chinese, Italian, Japanese, Portuguese, etc. We will explore cross-language and cross-domain robustness in future work.
>
> **Q1**: *“each phrase possesses a relatively complete and well-defined meaning” -> Will this approach be not generalizable to languages and domains where the availability of syntactic parsers is limited? Also is it feasible to annotate the whole training set?*
>
> **A3**: Please refer to our answer to W2. We would like to clarify that today’s syntactic parsers cover most common languages. For situations where a syntactic parser is unavailable, alternative methods may be utilized such as unsupervised syntactic parsing and unsupervised tokenization methods (*e.g.,* BPE, sentencepiece).
>
> As for the concern about annotating the whole training set. The Stanza parser we used only occupies about 100MB in disk (approximately 0.027B parameters). In fact, the syntactic parsing of the whole MiniPile dataset (5.7GB in disk) takes approximately **10 hours on 8 V100 GPUs**. Therefore, the cost of annotating the training set is relatively small compared to the cost of training the model.
>
> **Q2**: *“Incorporating high-frequency phrases can significantly increase the total number of phrases, leading to an extremely large candidate pool” -> Won’t the low-frequency phrases significantly increase the size of the candidate pool?*
>
> **A4**: Thanks for your insightful question! Low-frequency phrases do contribute to the size of the candidate pool. However, we find that high-frequency phrases have a much larger impact by contrast. Note that lexically identical phrases in different contexts are treated as different entries in the phrase pool. This means that a high-frequency phrase could potentially introduce tens of thousands, or even millions, of entries. In processing Wikipedia, we found that by removing just the **top** **1%** high-frequency phrases, we can reduce the total number of entries by **50%**. In contrast, the top **69%** of phrases with the lowest frequency account for only **15%** of the total number of entries.
>
> **Q3**: *Second paragraph under “Semantic similarity”: I felt lots of details were missing here to better understand the quality of phrases, and the feasibility of the proposed approach. The Appendix A do not provide all necessary details. Is this done on the pretraining corpus? What trivial constituents were dropped out and why (some examples would help)?*
>
> **A5**:
>
> - For the feasibility of the proposed approach, we believe you're concerned about the time cost of the entire preprocessing process. We processed the entire pretraining corpus (i.e., MiniPile and Wikipedia). This process (including syntactic parsing, phrase selection, and semantic matching) took approximately **24 hours on 8 V100 GPUs**. The overhead is small compared to the cost of training the model.
> - As for trivial constituents, we refer to constituents that are not semantically rich, such as conjunctions (*e.g., "and", "or"*), predeterminers (*e.g., "both", "all"*), wh-determiners (*e.g., "which"*), and wh-pronouns (*e.g., "who", "what"*).
> - We will add more details as mentioned above to the next version of this paper. Thanks for your suggestions.

---

> > ### Author Response · Authors · 2023-11-20
> >
> > **Q4:** *Sec 3.2.2: I found the explanation a bit confusing. Could you add an algorithm and/or an example demonstrating the algorithm?*
> >
> > **A6:**
> >
> > - Algorithm:
> >     1. Initialize the model with the initial generation paths determined by the heuristics.
> >     2. For each prefix p in the training data:
> >         1. Use the current model policy to retrieve the *k*-best phrases from the entire candidate pool.
> >         2. Choose the valid phrase, which is a continuation of the prefix in the ground truth, with the highest semantic matching score from these *k* phrases as the new target.
> >         3. If no valid phrase is found, retain the previous target.
> >     3. Update the model with the new targets.
> >     4. Repeat steps 2-3 periodically until the model's performance converges.
> > - Example:
> >     - Suppose we have a prefix p = *"Go right for the top when you"*. The ground truth for this prefix is *"Go right for the top when you want to make things happen"*. The initial target phrase determined by the heuristics might be *"want".*
> >     - In the iterative self-reinforcement process, we would first let the model retrieve the *k*-best phrases for the prefix from the entire candidate pool. Supposing that the *k*-best phrases are [*"want", "want to", "want to make things happen", "need", "can"*], only *"want", "want to",* and *"want to make things happen"* are considered as valid ones. If the model's semantic matching score is highest for "want to make things happen", we would update the target phrase for the prefix to this phrase. If none of the *k*-best phrases are valid, we will retain the previous target *"want"*.
> > - We will add the above algorithm and example to our next version of this paper.
> >
> > **Q5:** *“If no such phrase is found, we retain the previous target.” -> When would this occur? When the candidate pool is empty?*
> >
> > **A7:** It refers to the situation where none of the *k*-best phrases are valid. A valid phrase is one that matches the ground-truth continuation. For example, for the text *"Go right for the top when you want to make things happen"*, given the prefix *"Go right for the top when you"*, the valid phrases include *"want", "want to", "want to make", "want to make things"*, and *"want to make things happen"*. If the model retrieves phrases such as *"can", "need"*, or *"should"* that do not match the ground-truth continuation, these phrases are considered not valid.
> >
> > **Q6**: *“we also add the token vocabulary to our phrase table” -> Are they subword units? What is the vocabulary?*
> >
> > **A8**:*“the token vocabulary”* refers to the original token vocabulary of the Base LM. For the GPT2 series, some tokens in the vocabulary are subword units (GPT-2 employs BPE to construct its vocabulary).
> >
> > **Q7**: *“We train our model on the training set of MiniPile2 (Kaddour, 2023)” -> What is this dataset? Is it a pretraining set of finetuning set? How are they used during training? This dataset is discussed again in 5.2.*
> >
> > **A9:** The MiniPile dataset is a large-scale, diverse corpus that is curated for language model pretraining. It is a subset of the Pile dataset, which is a collection of 22 diverse text sources, including books, websites, and other types of text. The MiniPile dataset is designed to be representative of the full Pile, but at a smaller scale, making it more manageable for training purposes. MiniPile has official train/dev/test subsets. We use the training set of MiniPile for training our model, and the test set of MiniPile for evaluating the performance of open-ended text generation.
> >
> > **Q8**: *“Note that the sum of all possible paths can be computed efficiently using dynamic programming with time complexity O(n 2 ), where n represents the number of tokens in the text” -> Will this be limiting for long form outputs?*
> >
> > **A10**:
> >
> > - The process of scoring all possible paths consists of two steps:
> >     1. Computing the transition scores from any starting positions, which has a time complexity of *O(n)*, and
> >     2. Running a dynamic programming algorithm to compute the sum of all possible paths, which has a time complexity of *O(n^2)*.
> >
> >     The majority of the time cost lies in the first step. In the second step, there are **at most *n^2* scalar addition and multiplication operations**, which have a very small overhead.
> >
> >
> > **Q9**: *Table 1: Are the numbers for baselines taken from the respective papers or are they reproduced by the authors?*
> >
> > **A11**: The results presented in Table 1 are obtained from our runs using publicly available code, which can be found through the following links
> >
> > https://github.com/lucidrains/RETRO-pytorch
> > https://github.com/gmftbyGMFTBY/Copyisallyouneed
> >
> > **Q10**: *Sec 6: Results: This should be Sec 5.2.2?*
> >
> > **A12**: You are right. Thanks for pointing it out.

---

### Meta-Review · Area_Chair_nvCE · 2023-12-04

**Metareview:**

This paper is concerned with retrieval-augmented generation and proposes a novel approach that utilizes a large vocabulary, including variable-length phrases retrieved from similar contexts. While the motivation of this work is similar to that of Copy-Generator (Lan et al., 2023), the present work does not use a two-stage process—instead, it offers a more unified approach that removes the dependence on document retrieval. The reviewers generally found the approach to be interesting and novel, and its methodology is overall quite sound. The approach yields strong results on a variety of benchmarks from OpenBookQA and open-ended generation tasks. The authors addressed most of the reviewers' concerns during the discussion period, and in particular, added a human evaluation that confirmed the findings of the paper. Given this work's novelty and good results, I recommend accepting this submission.

**Justification For Why Not Higher Score:**

While the paper is generally solid, the reviewers were slightly concerned about the use of linguistic heuristics for phrase extraction, which somewhat reduces the generalizability of this work. While the authors' response ("phrase extraction is based on general linguistic principles like syntactic structure and semantic similarity") are quite reasonable, this reliance on linguistic heuristics would presumably reduce the appeal of this paper at a venue like ICLR. Therefore, I recommend a less high-profile form of presentation (e.g., a poster).

**Justification For Why Not Lower Score:**

All reviewers agreed that the paper makes significant contributions, while not having significant flaws. I found the responses generally convincing, as they addressed the main limitation of the initial submission (lack of human evaluation). Retrieval-augmented generation is a timely subject, and I believe this work would be of significant interest to the community.

---

### Decision · Program_Chairs · 2024-01-16

Accept (poster)